# PERFCODER: LARGE LANGUAGE MODELS FOR INTERPRETABLE CODE PERFORMANCE OPTIMIZATION

## ABSTRACT

Large language models (LLMs) have achieved remarkable progress in automatic code generation, yet their ability to produce high-performance code remains limited—a critical requirement in real-world software systems. We argue that current LLMs struggle not only due to data scarcity but, more importantly, because they lack supervision that guides interpretable and effective performance improvements. In this work, we introduce **PerfCoder**, a family of LLMs specifically designed to generate performance-enhanced code from source code via interpretable, customized optimizations. PerfCoder is fine-tuned on a curated collection of real-world optimization trajectories with human-readable annotations, and preference-aligned by reinforcement fine-tuning using runtime measurements, enabling it to propose input-specific improvement strategies and apply them directly without relying on iterative refinement. On the PIE code performance benchmark, PerfCoder surpasses all existing models in both runtime speedup and effective optimization rate, demonstrating that performance optimization cannot be achieved by scale alone but requires optimization stratetgy awareness. In addition, PerfCoder can generate interpretable feedback about the source code, which, when provided as input to a larger LLM in a planner-and-optimizer cooperative workflow, can further improve outcomes. Specifically, we elevate the performance of 32B models and GPT-5 to new levels on code optimization, substantially surpassing their original performance.

## 1 INTRODUCTION

Large language models (LLMs) such as Codex (Chen et al., 2021), GPT-4/5 (OpenAI et al., 2023), and Code Llama (Roziere et al., 2023) have substantially advanced automatic code generation, enabling natural language prompts to be translated into syntactically and semantically correct programs. Although these models excel at producing functionally correct code, they remain limited in optimizing code implementations for performance–an essential ability for building efficient, scalable software systems to meet strict latency and runtime requirements.

As illustrated in Figure 1, even advanced LLMs like ChatGPT[1] often generate transformations that appear plausible but degrade or fail to improve code runtime performance. This shortcoming arises because most models are trained in general-purpose corpora with little efficiency-related supervision (Shypula et al., 2024). Even when exposed to improved solutions, they typically lack the ability to explain or justify their code edits, but instead rely on opaque trial-and-error heuristics (Gao et al., 2024). Recent efforts have investigated data curation, fine-tuning schemes, and search-based inference-time scaling (Du et al., 2024; Huang et al., 2024a). Yet these approaches often remain black-box and non-interpretable, making them difficult to generalize and remain limited in performance.

In this paper, we introduce **PerfCoder**, a family of fine-tuned language models specifically targeting at interpretable and customized code performance optimizations. Given the input slow program, PerfCoder generates human-readable optimization strategies tailored to the program and applies them to the input context, allowing for code transformations in a reliable, transparent, and traceable way. Furthermore, PerfCoder does not rely on feedback or multi-step refinement (Huang et al., 2024a) during inference, and can be deployed as either a standalone code optimizer in a single step or as a planner to guide another (usually larger) model to optimize code performance in a cooperative procedure.

We enable PerfCoder's optimization capability through a fine-tuning procedure grounded in AI-synthesized code optimization trajectories from real-world code implementations. First, we recon-

---

[1] https://chatgpt.com/

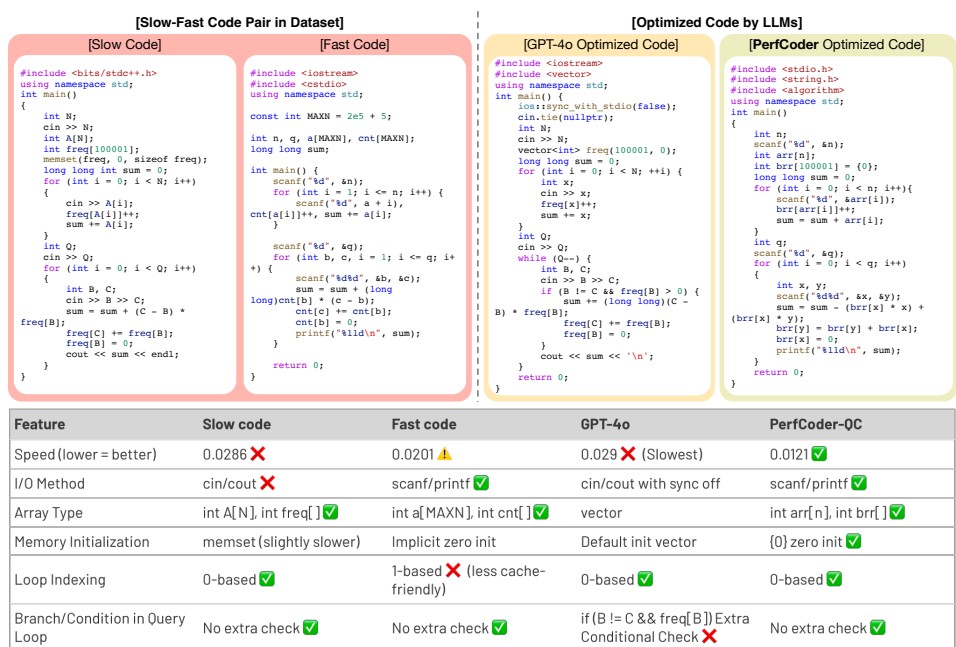

Figure 1: A real code optimization case of PerfCoder and ChatGPT.

struct the PIE dataset (Shypula et al., 2024), assembling an evaluation-aligned corpus of 30,649 slow–fast program pairs via endpoint selection. We then automatically extract optimization strategies from the code pairs, mapping them into core strategy primitives to provide automated supervision for structured reasoning. We propose sampling procedures to yield high quality data that is used to train PerfCoder into a single-step code optimizer which can generate both structured strategies and optimized code for the given input. we then introduce a reinforcement fine-tuning procedure to further incentivize the strategy generation ability of PerfCoder by fine-tuning it in a planner-and-optimizer cooperative framework using measured runtime as reward signals.

On the PIE benchmark, PerfCoder achieves substantial gains over existing baselines. A 7B version delivers a $2.50\times$ runtime speedup, surpassing both single-step methods (e.g., PIE-CodeLlama at $1.89\times$) and larger models such as Qwen2.5-Inst-32B ($1.50\times$). Its strategy outputs also generalize; when used to guide stronger LLMs in planner–optimizer mode, they provide significant additional improvements. Furthermore, RL fine-tuning with runtime feedback can align PerfCoder's strategy generation ability directly with code optimization outcomes. A small 1.5B PerfCoder can guide a 32B optimizer to achieve $3.03\times$ speedup, and guide GPT-5[2] to achieve $4.82\times$ speedup (a significant leap from $1.96\times$ speedup by GPT-5 alone without using PerfCoder as a planner).

In summary, PerfCoder establishes a practical and interpretable framework for code performance optimization. By combining strategy-aware supervision, balanced data curation, and RL preference alignment, it consistently outperforms a wide range of baselines, enhances larger models, and moves toward closing the gap between correctness and efficiency. To help support further research, we will publicly release our code, models, and the curated dataset at *Anonymous*.

## 2 METHOD

This section presents the design of **PerfCoder**. In contrast to prior work that focuses mainly on behavioral imitation or reinforcement by runtime metrics, PerfCoder learns through *structured strategy induction*. Given unoptimized code, it can either directly generate optimized code or act as a *planner* that guides an external (usually larger) optimizer model. Our fine-tuning procedure consists of two main stages, with an overview shown in Figure 2. First, we introduce a supervised fine-tuning (SFT) scheme to obtain PerfCoder Jr. which can generate optimization strategies followed by the corresponding code edits in a single-step mode. We describe our unified and automated data curation

---

[2]https://openai.com/index/introducing-gpt-5/

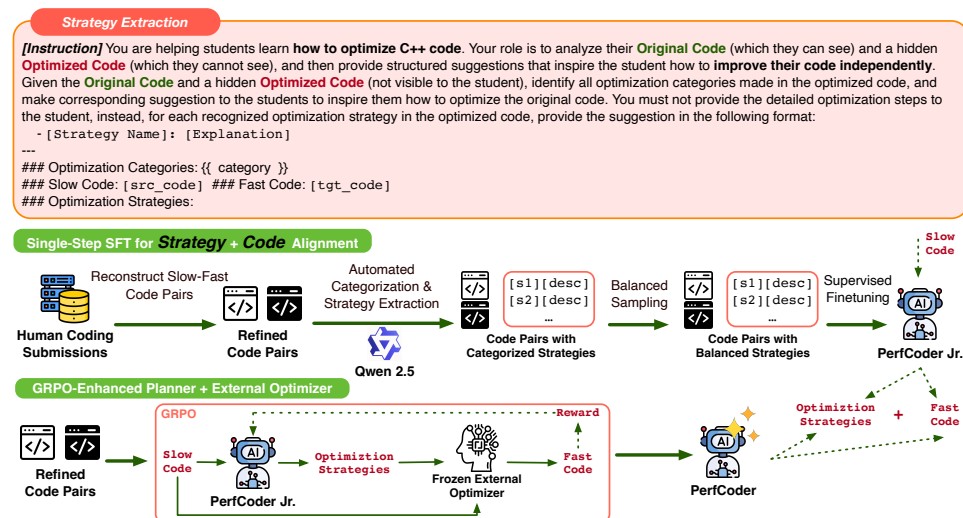

Figure 2: An illustration of our PerfCoder framework.

pipeline for SFT. Second, we introduce a reinforcement fine-tuning process where PerfCoder is fine-tuned into a stronger planner, in a planner-optimizer cooperative framework, to generate effective natural language code optimization strategies for optimizers to follow.

## 2.1 SINGLE-STEP CODE OPTIMIZATION MODE AND SUPERVISED FINE-TUNING

Unlike costly iterative self-refinement, in single-step mode, PerfCoder adopts a *single-step* format that can generate optimization strategies and optimized code in one autoregressive sequence (one LLM invocation). This design explicitly aligns *what* to optimize with *how* to implement it.

Consider a user $u$ solving a programming problem $p$. Let $x_{\text{slow}}^{(u,p)}$ denote a slow submission and $x_{\text{fast}}^{(u,p)}$ a corresponding faster solution. Each training instance additionally includes a natural-language instruction $\mathcal{I}$ and a set of extracted optimization strategies $\mathbf{s}^{(u,p)} = \{s_1, \ldots, s_k\}$ describing the transformations from $x_{\text{slow}}^{(u,p)}$ to $x_{\text{fast}}^{(u,p)}$. To serialize these elements, we introduce control tokens

$$\mathcal{V}_{\text{ctl}} = \{[\text{SUGG/}], [/\text{SUGG}], [\text{OPT/}], [/\text{OPT}]\},$$

which delimit the strategy and code spans. Given $(\mathcal{I}, x_{\text{slow}}^{(u,p)})$, the model $g_\phi$—a transformer-based language model parameterized by $\phi$—is trained to generate the structured sequence

$$y^{(u,p)} = [\text{SUGG/}] \, \mathbf{s}^{(u,p)} \, [/\text{SUGG}] \, [\text{OPT/}] \, x_{\text{fast}}^{(u,p)} \, [/\text{OPT}]. \tag{1}$$

Each strategy $s_i = (\text{name}_i, \text{desc}_i)$ consists of a canonical identifier (e.g., $\text{Loop Optimization}$) and a context-specific explanation of why the transformation improves $x_{\text{slow}}^{(u,p)}$. This serialization provides interpretable plans that connect abstract reasoning with concrete implementations.

The training objective is the standard causal language modeling loss. Let $P(\cdot \mid \cdot; \phi)$ denote the token distribution predicted by $g_\phi$. Then

$$\mathcal{L}_{\text{LM}} = - \sum_{t=1}^{|y^{(u,p)}|} \log P\Big(y_t^{(u,p)} \,\Big|\, y_{<t}^{(u,p)}, \mathcal{I}, x_{\text{slow}}^{(u,p)}; \phi\Big), \tag{2}$$

where $y_t^{(u,p)}$ is the $t$-th token of the target $y^{(u,p)}$ and $y_{<t}^{(u,p)}$ its prefix. Here, $\phi$ explicitly denotes the trainable parameters of the model.

This single-step design naturally supports two inference modes. In *plan+code mode*, decoding continues through the $[\text{OPT/}]$ span to produce optimized implementations directly. In *plan-only mode*, generation halts at $[/\text{SUGG}]$, yielding a human-interpretable strategy plan that can guide a stronger external LLM. This dual capability allows PerfCoder to function as either a self-contained optimizer or a lightweight planner, enhancing its flexibility and practical impact across real-world optimization scenarios.

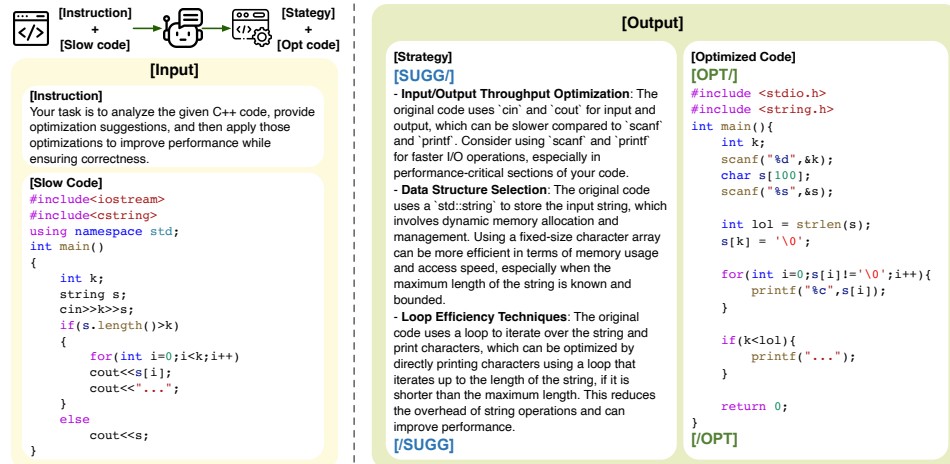

Figure 3: A real example of slow-fast code pair with optimization strategies. LLMs will learn from the output part to generate the strategy and the optimize code in a single-pass.

## 2.2 AUTOMATED CODE OPTIMIZATION STRATEGY SYNTHESIS

The strategies in the [SUGG/] segment (Eq. 1) are extracted automatically with a 32B open-source instruction-tuned model. By relying on an openly available model, the pipeline not only scales reliably but also ensures that every step can be reproduced by independent researchers. This automated design guarantees scalability, consistency, and reproducibility: instead of relying on human annotation, the system distills optimization knowledge directly from code trajectories. Formally, for each pair $(x_{\text{slow}}^{(u,p)}, x_{\text{fast}}^{(u,p)})$ in the curated dataset, the extractor $f_\theta$, parameterized by $\theta$, generates a corresponding set of strategies

$$\mathbf{s}^{(u,p)} = f_\theta\left(x_{\text{slow}}^{(u,p)}, x_{\text{fast}}^{(u,p)}\right), \tag{3}$$

which describe the transformations that turn the slower code into the faster one.

Each strategy is encoded as a tuple

$$s_i = (\texttt{name}_i, \texttt{desc}_i), \tag{4}$$

where $\texttt{name}_i$ is not arbitrary but drawn from a fixed set of fifteen canonical categories $\mathcal{C} = \{c_1, \ldots, c_{15}\}$, such as `Algorithm Design Optimization` or `Loop Efficiency Techniques`. The accompanying $\texttt{desc}_i$ is a natural-language explanation detailing why the technique benefits $x_{\text{slow}}^{(u,p)}$. This separation of a categorical "what" from a contextual "why" yields interpretable strategies that can generalize across diverse problems while remaining grounded in the local program context.

Mapping every strategy name to one of the fifteen categories provides structural regularization: it prevents frequent but superficial techniques from dominating training and ensures that rarer, long-tail strategies remain represented. The technical details of deduplication and category-guided strategy re-extraction are provided in the Appendix A.

Figure 3 illustrates outputs from this pipeline, showing how canonical category labels are paired with context-specific explanations to form interpretable and reusable optimization strategies.

## 2.3 DATASET RECONSTRUCTION AND BALANCED STRATEGY SAMPLING

To obtain reliable optimization strategies, it is essential to begin with high-quality code pairs. We build on the PIE dataset $\mathcal{D}_{\text{PIE}} = \{(x_{\text{slow}}^{(u,p)}, x_{\text{fast}}^{(u,p)})\}$, which provides abundant real-world trajectories of performance improvement. However, the raw dataset suffers from several shortcomings: optimization targets are often ambiguous due to intermediate submissions of real users, and the absolute quality of a user's final code may lag far behind problem-level best solutions. These issues introduce noise and weaken the learning signal. To overcome them, we reconstruct the dataset to impose clear optimization endpoints and then apply balanced sampling to reduce category bias.

Table 1: Dataset overview with symbolic representations. $\mathcal{D}_{\text{PIE}}$ is the original dataset. $\mathcal{D}_{\text{ref}}$ contains pairs where the optimized code is either the user's final or the global best submission. $\mathcal{D}_b$ is a strategy-category-balanced subset used for fine-tuning. "Cross-User Pairs" indicate cases where the optimized code was taken from a different user due to the user's poor performance.

| Dataset | Total Pairs | Description | Cross-User Pairs |
|---|---|---|---|
| $\mathcal{D}_{\text{PIE}}$ | 77,967 | Original PIE slow-fast pairs | – |
| $\mathcal{D}_{\text{ref}}$ | 30,649 | Reconstructed (slow vs. final/best) | 12,350 |
| $\mathcal{D}_b$ | 5,000 | Strategy-category-balanced subset of $\mathcal{D}_{\text{ref}}$ | 2,267 |

**Final-submission filtering.** For each user $u$ and problem $p$, we retain only the last submission as the optimization target, ensuring that every pair points toward a well-defined endpoint:

$$\mathcal{D}_{\text{ref}} = \left\{ (x_{\text{slow}}^{(u,p)}, x_{\text{final}}^{(u,p)}) \right\}. \tag{5}$$

**Global-best replacement.** Some final submissions remain significantly slower than the best-known solution for the same problem. To prevent the model from imitating under-optimized code, we replace such targets with the global best $x_{\text{best}}^{(p)}$. Formally, if the runtime $T(\cdot)$ satisfies

$$T\left(x_{\text{final}}^{(u,p)}\right) > 2 \cdot T\left(x_{\text{best}}^{(p)}\right), \tag{6}$$

then we substitute $x_{\text{fast}}^{(u,p)} \leftarrow x_{\text{best}}^{(p)}$. This correction grounds training in performance-competitive implementations rather than local improvements.

**Balanced strategy sampling.** Even after reconstruction, strategy frequencies are highly skewed: common strategies such as *loop unrolling* dominate, while rare yet more important strategies are under-represented. To counteract this, we assign each pair a rarity-weighted score $S^{(u,p)} = \frac{1}{k} \sum_{i=1}^{k} \frac{1}{f(s_i)}$, where $f(s_i)$ denotes the global frequency of strategy $s_i$. The pairs are ranked by $S^{(u,p)}$, and round-robin selection is applied to form a balanced subset $\mathcal{D}_b \subset \mathcal{D}_{\text{ref}}$, with $|\mathcal{D}_b| = 5000$, which preserves long-tail coverage and prevents the model from overfitting to the most frequent strategy categories. Please refer to Appendix C (Figure 5) for the distribution of code optimization strategy categories in the dataset before and after applying our category-balanced sampling procedure, with Table 4 providing the detailed explanation of each category.

## 2.4 Reinforcement Fine-Tuning in Planner Mode

**Planner mode.** While single-step supervised fine-tuning already provides substantial gains as we will show in Section 3, its effectiveness still depends heavily on the coding capacity of the base model, i.e., how well it can generate code following instructions, an ability usually tied to the model size. In contrast, generating effective optimization strategies for the given code does not necessarily require a large coding model. Motivated by this intuition, we further fine-tune PerfCoder with Group Relative Policy Optimization (GRPO), where PerfCoder serves as a smaller *planner* model that outputs optimization strategies only for the given code, while another larger external model serves as the *optimizer* to follow these proposed strategies.

As shown in Figure 2, during GRPO fine-tuning, PerfCoder operates in *planner* mode, i.e., decoding terminates at `[/SUGG]`, which is treated as an end-of-sequence token. The set of generated strategies (together with the slow input code) are passed as prompts into a larger external LLM, referred to as *optimizer*, which generates optimized code by applying these PerfCoder-generated strategies. We only fine-tune the smaller planner (PerfCoder) with GRPO while freezing the optimizer, using an end-to-end reward obtained from measured outputs.

**Reward design.** Let $T(\cdot)$ denote the measured runtime of a program. Given a slow input code $x_{\text{slow}}^{(u,p)}$ and optimizer-generated code $x_{\text{gen}}^{(u,p)}$, we define the speedup factor as

$$\Delta = \frac{T\left(x_{\text{slow}}^{(u,p)}\right)}{T\left(x_{\text{gen}}^{(u,p)}\right)}. \tag{7}$$

To encourage compilable and competitive output, reward is then assigned as

$$
R = \begin{cases} -\omega, & \text{if } x_{\text{gen}}^{(u,p)} \text{ fails to compile,} \\ -1, & \text{if } \Delta < 1 \text{ (slower than baseline),} \\ \Delta^2, & \text{if } \Delta \geq 1 \text{ (speedup achieved).} \end{cases} \tag{8}
$$

Here we set $\omega$ to 100 to severely penalize uncompilable or regressive outputs, while quadratically rewarding positive runtime gains. The quadratic scaling of $\Delta$ incentivizes the discovery and use of strategies that can produce not only valid but also significantly faster implementations.

Following GRPO training objective Shao et al. (2024), we perform relative comparisons within groups of end-to-end edits to calculate advantages. That is, for each input slow code $x_{\text{slow}}^{(u,p)}$, the planner is called multiple times to generate a group of (e.g., 4) different strategies, each of which separately guides the optimizer to generate a different piece of output code. Each output code is scored with the reward (a function of runtime speedup) mentioned above. Then, GRPO compares each output's reward relative to the group's average. We only fine-tune the PerfCoder planner with such reward signals while freezing the larger optimizer. Please refer to Appendix B for GRPO taining details.

Although PerfCoder can also produce strategies and code directly in single-step mode (Section 2.1), reinforcement fine-tuning of planner alone ensures that reward signals target strategy generation alone (which is the missing ability even in current large base models) instead of instruction following. Through this further alignment, strategies that generate compilable code with relatively higher speedup are reinforced and made more likely, while weaker or harmful ones are suppressed. However, the SFT of PerfCoder in single-step mode is necessary, since it has aligned meaningful strategy generation with code generation, providing a starting point for further reinforcement learning.

## 3 EXPERIMENTAL RESULTS

We evaluate PerfCoder on the PIE benchmark (Shypula et al., 2024), which consists of 978 unoptimized C++ programs drawn from 41 competitive programming problems. All experiments follow the PIE evaluation protocol and report the following three metrics:

*Speedup*, the primary performance indicator, is defined as Speedup $= \frac{t_{\text{slow}}}{t_{\text{fast}}}$, measuring how much faster the optimized code runs relative to the original. Each optimized program is evaluated on 20 test cases. Only those that pass all test cases are considered correct; otherwise, the program is treated as incorrect and assigned a speedup of 1. Additionally, we report *Effective Optimization* rate, the percentage of generated programs that are both correct and achieve at least 1.1× speedup, and *Code Accuracy*, the percentage of programs passing all functional test cases.

While code accuracy evaluates functional correctness, it does not imply meaningful performance improvement. A model can achieve high accuracy yet produce code that is inefficient. Effective optimization, by contrast, ensures both correctness and speedup, making it a more practical metric for real-world deployment.

**Baselines.** We compare PerfCoder against a broad range of instruction-tuned and open-source baselines. This includes smaller models such as CodeLlama-7B (Roziere et al., 2023) and Olympic-Coder (Face, 2025), as well as larger 32B-scale models like Qwen2.5-Inst (Team, 2024a), Qwen2.5-Coder-Inst (Team, 2024b), and DeepSeek-R1-Distill-Qwen (Team, 2025). We also include models fine-tuned on high-quality PIE subsets, including PIE-CodeLlama and PIE-Qwen2.5-Coder.

To isolate the effect of our balanced dataset and strategy supervision, we fine-tune each model under identical conditions where applicable.

We evaluate models under two distinct inference modes. In the single-step setting, models are prompted with the unoptimized (slow) code and directly generate the optimized version, including any embedded strategies. In the two-step setting (include GRPO), models first generate a set of optimization strategies, and are then re-prompted to produce optimized code conditioned on those

Table 2: Main Results. "-HQ" indicates LLMs fine-tuned on the high-quality datasets from the PIE paper. Model name in bold represent our proposed approach, fine-tuned on the category-balanced dataset constructed using our sampling method.

| Method | Model Size | Inference Steps | Speedup | Effective Optimization | Code Accuracy |
|---|---|---|---|---|---|
| GPT-4 | - | Single-Step | 1.32× | 26.99% | 63.09% |
| GPT-5 | - | Single-Step | 1.96× | **53.25%** | **93.66%** |
| CodeLlama-Inst | 7B | Single-Step | 1.04× | 3.17% | 30.27% |
| Olympic-Coder | 7B | Single-Step | 1.08× | 2.56% | 18.20% |
| Qwen2.5-Inst | 32B | Single-Step | 1.50× | 26.69% | 72.29% |
| Qwen2.5-Coder-Inst | 32B | Single-Step | 1.39× | 22.90% | 68.40% |
| DeepSeek-R1-Distill-Qwen | 32B | Single-Step | 1.23× | 11.25% | 38.55% |
| PIE-CodeLlama-HQ | 7B | Single-Step | 1.73× | 26.58% | 41.41% |
| PIE-Qwen2.5-Coder-HQ | 7B | Single-Step | 1.98× | 32.62% | 42.64% |
| **PerfCoder-CL** | 7B | Single-Step | 1.94× | 21.47% | 31.60% |
| **PerfCoder-QC** | 1.5B | Single-Step | 1.81× | 17.18% | 20.35% |
| **PerfCoder-QC** | 7B | Single-Step | **2.50×** | 33.13% | 43.46% |
| Effi-Learner w/o history | 32B | Five-Rounds | 1.47× | 26.01% | 64.21% |
| Effi-Learner w/ history | 32B | Five-Rounds | 1.54× | 26.28% | 64.72% |
| Qwen2.5-Coder-Inst | 32B | Two-Step | 1.38× | 20.86% | 72.29% |
| Qwen2.5-Inst | 32B | Two-Step | 1.32× | 20.76% | 80.37% |
| **PerfCoder-QC**+Qwen2.5-Coder-Inst | 7B+32B | Two-Step | **2.26×** | **44.89%** | 61.66% |
| **PerfCoder-QC**+Qwen2.5-Inst | 1.5B+32B | Two-Step | 2.54× | 43.05% | 62.27% |
| **PerfCoder-QC**+Qwen2.5-Inst | 7B+32B | Two-Step | 2.52× | 43.56% | 60.63% |
| **PerfCoder-QC**+Qwen2.5-Inst | 1.5B+32B | Two-Step with GRPO | **3.03×** | 48.06% | 59.00% |
| **PerfCoder-QC**+GPT-5 | 1.5B+GPT-5 | Two-Step with GRPO | **4.82×** | **79.86%** | **97.95%** |

strategies. This two-step procedure is tested in two configurations: (1) using strategies generated by the model itself, and (2) using strategies provided by PerfCoder. This setup allows us to assess both the internal strategy reasoning capabilities of large LLMs and the transferability of PerfCoder's explicitly trained strategies.

Additionally, we reproduce Effi-Learner (Huang et al., 2024a) in the C++ environment. Because the original relies on Python-specific profilers (`line_profiler`, `memory_profiler`), we replace them with `gcov` and end-to-end runtime, ensuring a fair comparison on PIE. The vanilla system queries the LLM only with the previous round's code; we denote this as `Effi-Learner w/o history` in Table 2. To test the role of context, we also evaluate `Effi-Learner w/ history`, which conditions on the full conversation history, allowing the model to build on accumulated reasoning and prior generations.

**Infrastructure and Hyperparameter.** All experiments are conducted on a single-node server with 4× NVIDIA V100 32GB GPUs, except for GRPO training, which requires an additional 8× NVIDIA V100 32GB GPUs. We fine-tune all models for 2 epochs with a batch size of 64 and a learning rate of $2 \times 10^{-5}$, following the protocol described in the PIE paper. Decoding is performed using greedy search. For GRPO, we fine-tune PerfCoder for 1 epoch with 4 generations per sample and employ Qwen-2.5-32B-Inst as the optimizer, while keeping all other settings unchanged. To evaluate PerfCoder's planning ability, we use the GRPO-finetuned PerfCoder to guide GPT-5 in code optimization.

## 3.1 RESULTS ANALYSIS

Table 2 presents a comprehensive comparison across all evaluated models under both single-step and two-step inference settings. Our analysis yields five key findings.

**(1) PerfCoder achieves state-of-the-art single-step performance.** Both variants of PerfCoder outperform all direct optimization baselines in speedup. PerfCoder-CL, based on CodeLlama-7B, achieves a 1.94× speedup, exceeding PIE-CodeLlama-HQ (1.73×) despite PIE-CodeLlama being fine-tuned with high-quality data distilled from GPT-3.5 (OpenAI, 2023). PerfCoder-QC, based on Qwen2.5-Coder-7B, achieves a 2.50× speedup and 33.13% effective optimization—surpassing PIE-Qwen2.5-Coder-HQ (1.98×, 32.62%) and even the much larger 32B Effi-Learner (1.54×, 26.28%). Notably, PerfCoder-QC-7B also outperforms GPT-4 (1.32×, 26.99%) and GPT-5 (1.96×, 53.25%) in runtime speedup, despite being smaller and open-source. These results validate the effectiveness of our strategy-aware, single-step framework and demonstrate that interpretable, optimization-trajectory supervision can outperform both scale and proprietary pretraining.

**(2) PerfCoder strategies generalize to larger models.** When used in a two-step setup to guide stronger models, PerfCoder's strategies produce substantial performance gains. Qwen2.5-Inst improves from 1.32× to 2.52× speedup and from 20.76% to 43.56% effective optimization when guided

Table 3: Ablation study results. We test the performance of fine-tuning LLMs without optimization strategy or category balancing.

| Method | Speedup | Effective Optimization | Code Accuracy |
|---|---|---|---|
| **PerfCoder-QC** | **2.50×** | **33.13%** | 43.46% |
| w/o Strategy | 2.11× | 32.62% | 63.08% |
| w/o Balancing | 2.09× | 20.76% | 33.64% |

by PerfCoder-QC. Qwen2.5-Coder-Inst shows a similar trend, improving from 1.38× to 2.26× and from 20.86% to 44.89%. These results demonstrate that PerfCoder strategies are highly transferable and provide effective, interpretable guidance for general-purpose LLMs. These results demonstrate that PerfCoder's strategies are not only interpretable but also highly transferable—allowing general-purpose models to benefit from targeted optimization knowledge even without additional fine-tuning.

**(3) Strong optimization guidance does not require large models.** Even small models can serve as effective optimization planners. For example, PerfCoder-QC-1.5B, without any preference learning, already produces strategies that enable a 32B model to reach performance comparable to that achieved with PerfCoder-QC-7B as the planner. With just one epoch of GRPO fine-tuning, the 1.5B variant provides even stronger guidance, allowing the 32B model to achieve a 3.03× speedup—surpassing all baselines.

**(4) Strategy generation without supervision can be harmful.** When large models such as Qwen2.5-Inst and Qwen2.5-Coder-Inst are prompted to generate and follow their own optimization strategies, performance drops substantially—reaching only 1.32× and 1.38× speedup, respectively, which is worse than their single-step baselines. This degradation arises not because strategy conditioning is ineffective, but because these general-purpose models lack optimization-specific supervision and thus often produce vague, incorrect, or misleading strategies. Unlike PerfCoder, which is explicitly fine-tuned to generate actionable and interpretable strategies grounded in real transformations, these models are not strategy-aware and may inadvertently misguide the optimization process. This highlights the importance of supervised strategy modeling and validates PerfCoder's design for delivering reliable planning signals (Section 2).

**(5) Code correctness does not guarantee optimization.** Correctness is necessary but insufficient for optimization. For example, Qwen2.5-Inst attains the highest code accuracy (80.37%) in the two-step setting, yet only 20.76% of its outputs yield effective optimizations (at least 1.1× faster). In contrast, PerfCoder-QC reaches 33.13% effective optimizations despite a lower accuracy of 43.46%, showing that strategy-aware learning favors performance gains over merely valid code. This stems from our curated dataset (Section 2), which aligns targets with user-final or globally optimal solutions, encouraging models to seek impactful transformations. In practice—where runtime and throughput dominate—*effective optimization* is thus a more actionable metric than correctness alone.

## 3.2 ABLATION STUDY

We ablate the two key components of PerfCoder: category-balanced sampling and strategy-aware supervision.

First, removing strategy supervision—training the model directly on optimized code without revealing the underlying intent—reduces speedup. While these models often generate functionally correct code, they struggle to internalize performance-improving transformations. This is because direct imitation encourages surface-level learning of final outputs, biasing the model toward correctness rather than efficiency. In contrast, PerfCoder's interpretable and customizable strategy supervision explicitly teaches the model why and how to optimize, resulting in higher speedup.

Second, removing category-balancing reduces the model's exposure to rare but impactful strategies. This leads to an overemphasis on frequent patterns and degrades the model's ability to generalize beyond commonly seen optimizations.

These findings reinforce our central claim: effective optimization, not just correctness, is the proper objective for performance-critical code generation. Strategy-aware learning provides the most direct and interpretable signal for achieving this goal.

## 4 RELATED WORK

To assist software engineers in completing coding tasks more efficiently, LLMs for code generation have rapidly progressed, from generating simple code snippets (Li et al., 2024; Peng et al., 2024; Zhuo et al., 2024) to supporting repository-level code generation (Jimenez et al., 2023; Zhang et al., 2023; Liu et al., 2023; Ding et al., 2023). Concurrently, a growing body of research focuses on optimizing both original and generated code from various perspectives, including bug fixing (Jin et al., 2023; Xia et al., 2023; Dinh et al., 2023; Xia & Zhang, 2024; Liu et al., 2024), security enhancement (Berabi et al., 2024; Ahmad et al., 2024; Wu et al., 2023; Pearce et al., 2023), and performance improvement (Madaan et al., 2023; Du et al., 2024; Waghjale et al., 2024; Huang et al., 2024b; Rosas et al., 2024; Cummins et al., 2025; Niu et al., 2024; Coignion et al., 2024). Among these, performance optimization aims to enhance execution speed, reduce memory consumption, and improve energy efficiency—critical factors for real-time applications, edge deployment, and cloud cost reduction. As such, generating high-performance code is essential for enabling scalable and efficient intelligent systems in real-world scenarios.

Achieving such performance gains has motivated a line of work on improving the quality and efficiency of LLM-generated code during both training and inference. To this end, various search algorithms have been employed during training (Gao et al., 2024; Wang et al., 2024; Nichols et al., 2024; Duan et al., 2023; Ishida et al., 2024). Despite their effectiveness, these search-based and reinforcement learning methods are often computationally intensive and slow. Recent advances explore alternatives such as self-refinement (Du et al., 2024; Waghjale et al., 2024) and agentic approaches (Huang et al., 2024a; Chen et al., 2024a), which aim to improve performance but incur high token costs during inference due to multi-round generation.

A more direct and efficient strategy involves fine-tuning on paired examples of inefficient and optimized code, allowing models to learn performance-oriented transformations. Several recent efforts have explored different strategies to improve the performance of LLM-generated code. (Chen et al., 2024b) formulates the task as a Seq2Seq learning problem focused on generating optimized code patches, while (Ma et al., 2024) applies contrastive learning and instruction tuning to improve code quality based on problem descriptions. Both approaches require additional information or specialized models, which may limit their applicability. (Taneja et al., 2025) addresses the specific challenge of generating vectorizable code, but its technique lacks generalizability. In contrast, (Huang et al., 2024c; Madaan et al., 2023) highlight the importance of high-quality training datasets for enhancing model performance, though their inference processes lack interpretability. Inspired by these insights, we propose an intuitive and effective data construction and fine-tuning method for LLM-based code optimization, enabling an interpretable and customizable optimization process with improved speedup.

## 5 CONCLUSION

This work addresses the challenge of optimizing LLM-generated code, a critical step toward efficient and scalable systems. We introduce **PerfCoder**, a strategy-driven model that improves performance by learning and applying human-readable optimization strategies. Trained on a balanced, strategy-annotated dataset of real-world C++ optimizations, PerfCoder achieves notable runtime gains without iterative refinement or heavy external tooling. Experiments show that PerfCoder not only outperforms baselines in runtime speedup and effective optimization, but also provides interpretable strategies that guide larger LLMs more effectively. While our current extractor (Qwen2.5-32B-Inst) may limit strategy quality compared to frontier models, PerfCoder establishes a practical and reproducible foundation for strategy-aware optimization. Future work will explore stronger extractors, multi-language extensions, and hardware-aware tuning to further close the gap between code generation accuracy and execution efficiency.

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

## A  STRATEGY DEDUPLICATION AND CATEGORIZATION

To obtain the 15 categories used in our method, we perform the following two steps to consolidate and structure the extracted strategies. The resulting taxonomy is summarized in Table 4. An example is illustrated in Figure 4.

> **Direct Strategy Generation**
>
> **[Instruction]** You are helping students learn C++ code optimization. Given an **Original Code** submitted by a student and a corresponding **Optimized Code** (which the student cannot see), your task is to provide clear and structured optimization suggestions. Your goal is to guide the student in improving their **Original Code** step by step. The suggestions should be actionable so that, by following them, the student can independently transform their code into an optimized version—without seeing the **Optimized Code**. List all optimization strategies applied in the **Optimized Code**. Provide each strategy in this format:
>  - [Strategy Name]: [Explanation]
> Each explanation should clearly describe what needs improvement in the **Original Code**, why the optimization is beneficial, and how the student should modify their code to implement the optimization.
> ---
> ### Example 1: [src_code:example_1] + [tgt_code:example_1] + {{ strategy }}
> ### Example 2: [src_code:example_2] + [tgt_code:example_2] + {{ strategy }}
> ### Now analyze this pair: [src_code] + [tgt_code]
> #### Optimization Suggestions:

> **Categorization**
>
> **[Instruction]** You are an expert AI classifier specializing in code optimization strategies. Strictly follow these instructions:
> **Task**: 1. Analyze the provided optimization strategy. 2. Match it to exactly one category from the given list. 3. Return ONLY the category name or "N/A". 4. No explanations, disclaimers, or formatting.
> **Rules**: 1. If no clear match, immediately return "N/A". 2. Never modify or combine category names 3. Never infer beyond provided information 4. Response must be exactly the name of the category or "N/A".
> ---
> ### Categories (EXACT MATCH REQUIRED): {{ category }}
> ### Strategy to Classify: {{ strategy }}
> ### Classification Output:

Figure 4: An illustration of strategy deduplication and categorization.

**(1) Direct Extraction.** For each pair $(x_{\text{slow}}^{(u,p)}, x_{\text{fast}}^{(u,p)})$ in the curated dataset, we prompt the extractor $f_\theta$ to generate the corresponding optimization strategies:

$$\mathbf{s}^{(u,p)} = f_\theta\left(x_{\text{slow}}^{(u,p)}, x_{\text{fast}}^{(u,p)}\right). \tag{9}$$

The model outputs strategies in the structured tuple format $s_i = (\texttt{name}_i, \texttt{desc}_i)$, ensuring that each optimization is described both by a high-level technique label and by a contextual rationale tied to the input program.

**(2) Deduplication and Categorization.** Since lexical variation often leads to redundant expressions of the same optimization, we normalize strategy names to construct a unique set $\mathcal{S}_{\text{uniq}}$, yielding 60,650 distinct entries. To impose structure, we define a taxonomy of categories $\mathcal{C} = \{c_1, \ldots, c_{15}\}$ by manually inspecting 1,000 random names and then automatically classifying the remainder:

$$\text{Classify}(\texttt{name}_i) \ \rightarrow \ c_j \in \mathcal{C}. \tag{10}$$

This taxonomy supports filtering, balancing, and interpretability. Importantly, over 90.27% of extracted strategies align with the defined categories, demonstrating both the coverage and the robustness of the categorization.

## B GRPO TRAINING

Here we introduce the fine-tuning of PerfCoder using GRPO.

Let $\pi_\phi$ denote PerfCoder's policy over strategy sequences in planner mode. During training, for each slow program $x_{\text{slow}}^{(u,p)}$ with instruction $\mathcal{I}$, we sample a *set* of candidate strategies from the old policy:

$$\{\mathbf{s}_1, \ldots, \mathbf{s}_G\} \ \sim \ \pi_{\phi_{\text{old}}}(\cdot \mid \mathcal{I}, x_{\text{slow}}^{(u,p)}),$$

where $G$ is the number of samples. Each set of strategies $\mathbf{s}_i$ is passed to the optimizer, which produces optimized code $x_{\text{gen},i}^{(u,p)}$, and a reward $R(\mathbf{s}_i)$ is computed according to compilation and speedup (Section 2.4, reward design).

To stabilize training, rewards are normalized within the sampled group:

$$A_i \ = \ \frac{R(\mathbf{s}_i) - \text{mean}(\{R(\mathbf{s}_j)\}_{j=1}^G)}{\text{std}(\{R(\mathbf{s}_j)\}_{j=1}^G)}. \tag{11}$$

The optimization then follows a surrogate with group-relative advantage and KL regularization:

$$\max_\phi \ \mathbb{E}_{\mathcal{I}, x_{\text{slow}}^{(u,p)}, \mathbf{s}_i}\left[ \min\left(\rho_i(\phi)A_i, \ \text{clip}(\rho_i(\phi), 1-\varepsilon, 1+\varepsilon)A_i\right) \ - \ \beta\, D_{\text{KL}}\left(\pi_\phi \,\|\, \pi_{\text{ref}}\right)\right], \tag{12}$$

where
$$\rho_i(\phi) = \frac{\pi_\phi(\mathbf{s}_i \mid \mathcal{I}, x_{\text{slow}}^{(u,p)})}{\pi_{\phi_{\text{old}}}(\mathbf{s}_i \mid \mathcal{I}, x_{\text{slow}}^{(u,p)})},$$
$\varepsilon$ is the clipping parameter, and $\beta$ controls the KL penalty relative to a fixed reference policy $\pi_{\text{ref}}$.

Thus, for each slow program, PerfCoder generates a *set of strategies* (e.g., 4 in our experiments), receives group-relative rewards from the optimizer, and updates its policy so that strategies leading to higher speedups are increasingly favored.

## C  STRATEGY CATEGORY DISTRIBUTION

Figure 5 illustrates the distribution of optimization strategy categories in the dataset before and after applying our category-balanced sampling procedure. Table 4 give the detailed explanation of each category.

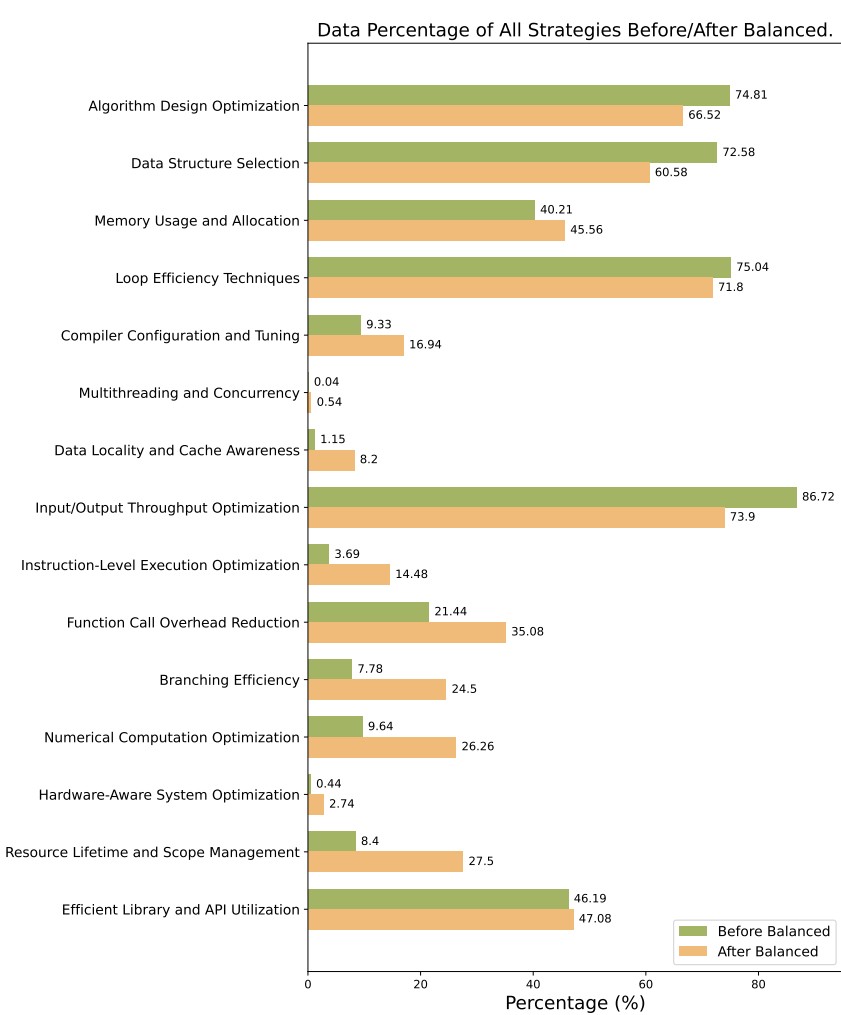

Figure 5: An illustration of data percentage of strategies in the training data before or after balanced sampling.

Prior to balancing, the dataset exhibits a strong skewness toward a few dominant strategy types. Categories such as *Input/Output Throughput Optimization* and *Loop Efficiency Techniques* account for the vast majority of samples—over 86% and 75%, respectively. In contrast, several meaningful yet underrepresented categories—such as *Multithreading and Concurrency*, *Data Locality and Cache*

*Awareness*, and *Hardware-Aware System Optimization*—appear in less than 1% of training pairs. This heavy imbalance limits the model's exposure to diverse optimization behaviors, biasing it toward over-represented patterns and reducing its capacity to generalize.

Table 4: Categories of Code Optimization Techniques

| Optimization Category | Description |
|---|---|
| Algorithm Design Optimization | Choosing or improving algorithms to make the program faster, more efficient, or simpler, etc. |
| Data Structure Selection | Using the right data structures for better performance, memory use, search speed, etc. |
| Memory Usage and Allocation | Managing how memory is allocated and accessed to reduce waste, improve speed, avoid fragmentation, etc. |
| Loop Efficiency Techniques | Optimizing loops to run fewer times, faster, or more efficiently with things like unrolling, breaking early, etc. |
| Compiler Configuration and Tuning | Using compiler flags or settings to let the compiler optimize the code automatically—like inlining, vectorizing, etc. |
| Multithreading and Concurrency | Running code in parallel using threads, tasks, or async techniques to make better use of CPU time, etc. |
| Data Locality and Cache Awareness | Organizing data in memory to take advantage of CPU caching and reduce access time, cache misses, etc. |
| Input/Output Throughput Optimization | Speeding up file, network, or console input/output through buffering, batching, async I/O, etc. |
| Instruction-Level Execution Optimization | Making use of low-level CPU capabilities like SIMD, pipelining, instruction reordering, etc. |
| Function Call Overhead Reduction | Reducing the cost of function calls by inlining, simplifying call chains, avoiding deep stacks, etc. |
| Branching Efficiency | Making conditionals faster by simplifying logic, reducing unpredictable branches, avoiding nested ifs, etc. |
| Numerical Computation Optimization | Making math-heavy code faster with better formulas, approximations, or hardware-accelerated operations, etc. |
| Hardware-Aware System Optimization | Tuning code for specific hardware features like CPU cores, vector units, cache size, memory bandwidth, etc. |
| Resource Lifetime and Scope Management | Managing the lifespan and ownership of resources like memory, files, threads to avoid leaks, race conditions, etc. |
| Efficient Library and API Utilization | Using well-optimized libraries, built-in functions, or system APIs instead of writing everything from scratch, etc. |

After applying our balancing method (described in Section 2), the long-tail categories are significantly upsampled, while the most frequent ones are proportionally reduced. For example, the frequency of the *Multithreading and Concurrency* category increases from 0.04% to 0.54%, and *Data Locality and Cache Awareness* increases from 1.15% to 8.2%. Meanwhile, the share of *Input/Output Throughput Optimization* decreases from 86.72% to 73.9%, preserving its presence but reducing its dominance.

This rebalancing procedure encourages the model to learn from a broader spectrum of optimization strategies. By promoting rare but impactful patterns, the balanced dataset enables better generalization and more robust performance—particularly on less common yet industrially relevant optimization scenarios. As evidenced in our ablation results, this leads to consistent improvements in effective optimization, even when overall code accuracy remains unchanged.

## D    TRANSFERABILITY TO OTHER BENCHMARKS

Table 5: Experimental results. We further fine-tune our model and PIE-Qwen2.5-Coder-HQ on a curated subset of PolyBenchC and evaluate their performance alongside other selected baselines.

| Method | Model Size | Inference Steps | Speedup | Effective Optimization | Code Accuracy |
|---|---|---|---|---|---|
| Qwen2.5-32B-Inst | 32B | Single-Step | 1.027× | 12.5% | 75.0% |
| PIE-Qwen2.5-Coder-HQ | 7B | Single-Step | 1.016× | 12.5% | 12.5% |
| **PerfCoder-QC** | 7B | Single-Step | **1.053×** | **25.0%** | 50.0% |

### D.1 DATA COLLECTION

To evaluate the transferability of PerfCoder beyond the PIE dataset, we construct a small auxiliary benchmark using the PolyBenchC suite (Pouchet, 2012; 2016). PolyBenchC consists of 30 loop-dominated numerical kernels characterized by static control flow, drawn from domains such as linear algebra, signal processing, dynamic programming, and scientific simulations.

We curate this benchmark in a two-stage process. First, for each function, we prompt several instruction-tuned language models—including CodeLlama-7B-Inst, CodeLlama-13B-Inst, and LLaMA3.3-Inst (Touvron et al., 2024)—with transformation-specific instructions targeting classic loop optimizations. These include loop unrolling (by factors of 2, 4, and 8), loop tiling, loop fusion, loop fission, operator strength reduction, and cache locality enhancement. Each prompt requests an optimized version of the given function using the specified transformation technique. This step results in a total of 1,620 generated code samples across model variants and prompt variations.

In the second stage, we filter the generated outputs to ensure quality and relevance. Specifically, we discard samples that either (i) fail to compile, (ii) exhibit no structural transformation compared to the original code, or (iii) do not yield any runtime performance gain when evaluated on an Intel Xeon server using `gcc` with `-O3` and `time` profiling. After filtering, we retain 185 unique and non-trivial optimized code instances that exhibit at least one interpretable transformation and measurable performance improvement over the baseline.

### D.2 EXPERIMENTAL SETTINGS

To evaluate the transferability of PerfCoder to new performance-critical domains, we conduct experiments on the PolyBenchC benchmark—a suite of loop-intensive scientific kernels commonly used in compiler and optimization research.

We randomly select 22 kernels from PolyBenchC for fine-tuning and use the remaining 8 kernels for evaluation. From the selected training set, we collect all available slow-fast pairs, yielding 141 training examples. Fine-tuning is performed for a single epoch using a learning rate of $1 \times 10^{-5}$ and a batch size of 4.

We apply this setup to fine-tune both **PerfCoder-QC** and the PIE baseline model (**PIE-Qwen2.5-Coder-HQ**) using the curated PolyBench training subset. Their performance is then evaluated on the held-out test kernels, alongside general-purpose LLMs in the single-step inference mode. The full results are reported in Table 5.

### D.3 EXPERIMENTAL ANALYSIS

Table 5 presents the evaluation results on the held-out PolyBenchC kernels. Among all models tested, PerfCoder-QC achieves the strongest transfer performance, yielding a speedup of $1.053\times$ and an effective optimization rate of 25.0%. In contrast, the baseline PIE-Qwen2.5-Coder-HQ, which lacks strategy-aware training and was fine-tuned on a high-quality subset of PIE using output-only supervision, achieves only $1.016\times$ speedup and 12.5% effective optimization—matching the score of Qwen2.5-32B-Inst, a significantly larger model (32B vs. 7B).

These results reinforce a core insight: explicit strategy modeling is more effective than mimicking optimized code alone. PerfCoder's use of interpretable, context-specific optimization strategies—paired with a category-balanced training set—enables it to generalize more robustly to structurally distinct tasks, such as numerical kernel optimization in PolyBenchC. Unlike code-only fine-tuning, strategy-guided supervision focuses the model's learning on *why* and *how* specific transformations yield performance gains, facilitating transfer to new domains.

Additionally, the experiment supports our earlier claim that effective optimization is a more meaningful metric than code accuracy in performance-critical scenarios. For example, Qwen2.5-Inst achieves the highest code accuracy on this benchmark (75.0%), yet only 12.5% of its outputs meet the threshold for effective optimization. Meanwhile, PerfCoder-QC, despite a lower code accuracy (50.0%), produces twice as many successful speedups. This reflects a known limitation of training on optimized code alone: the model may overfit to syntactic correctness without learning performance-centric transformations.

Figure 6: A real example from the PIE testset. Code segments highlighted in red fail to compile or do not pass all test cases, while those in green are functionally correct. The green numbers indicate the corresponding speedup. The rightmost boxes in each row show the optimization strategies proposed by PerfCoder-QC and Qwen2.5-32B-Inst (in a two-step setting), respectively.

# E  CASE STUDY

Figure 6 presents a real example from the PIE benchmark, highlighting the performance impact of strategy-aware optimization across multiple models and inference modes.

The original slow submission contains several inefficiencies, such as dynamic memory allocation via `std::vector`, slow C++ I/O using `cin`/`cout`, and redundant header files. The manually optimized reference version improves stability but yields only a moderate 2.20× speedup.

**PerfCoder-QC**, trained with strategy-aware supervision, applies three concrete strategies: (1) replacing I/O with `scanf`/`printf`, (2) using fixed-size arrays in place of vectors, and (3) optimizing loop bounds with `min()`. These result in a 3.75× speedup, demonstrating effective performance-oriented transformation.

**Qwen2.5-32B-Inst**, when used without external guidance, produces mostly stylistic edits—such as removing unused headers and variables—that yield only minor runtime improvement (1.055×).

However, when Qwen2.5-32B-Inst is guided by strategies extracted by PerfCoder-QC in a two-step inference setup, it achieves a dramatic 11.78× speedup. This not only outperforms all other models but also underscores the benefit of modular, interpretable optimization guidance.

Overall, this case study reinforces our core insight: strategy-aware supervision produces more meaningful and transferable optimization behaviors than code imitation alone, especially when paired with instruction-following models in collaborative settings.

## F    LLM USAGE

This paper uses a Large Language Model (LLM) only to polish English writing, including grammar, clarity, and style. All ideas, methods, and results are entirely authored by the researchers.

