# OpenReview forum: "PerfCoder: Large Language Models for Interpretable Code Performance Optimization"
_ICLR.cc/2026/Conference — ICLR 2026 Conference Withdrawn Submission_

### Official Review · Reviewer_ouK2 · 2025-10-25

**Soundness:** 2
**Presentation:** 2
**Contribution:** 2
**Rating:** 2
**Confidence:** 4

**Summary:**

This paper introduces **PerfCoder** to address the gap between functionally correct code generation and high-performance code optimization. The authors argue that existing LLMs fail at performance tasks due to a lack of supervision that connects optimizations to interpretable, human-readable strategies. They fine-tune PerfCoder under a two-stage fine-tuning process that first supervised fine-tune on a PIE restructure, and then using GRPO for futher refinement. Experiments on the PIE benchmark show that PerfCoder (7B) outperforms all single-step baselines in terms of runtime speedup, demonstrating the effectiveness and transferability of its learned strategies.

**Strengths:**

1. **Well-Defined Problem:** The paper tackles a critical and high-value limitation of current code-generation models. Moving from functional correctness to performance optimization is a necessary next step for LLMs to be truly useful in production software engineering.
2. **Small Model as Planer:** The insight that a small, specialized model can generate high-quality, interpretable strategies to guide a large, general-purpose optimizer model is impressive.

**Weaknesses:**

1. **Generalizability and Benchmark Limitations:** The models are trained on a reconstructed PIE dataset and evaluated on the original PIE benchmark, which may favor a particular subset of benchmark but not representative of real-world code. The transferability study to PolyBenchC is a good addition, while the resulting speedup is modest (1.053), suggesting the learned strategies may not generalize well to other domains without further domain-specific fine-tuning.
2. **Reliance on Oracle Training Data:** The "Global-best replacement" strategy requires access to the best-known solution for a given problem to create the training data. This strong oracle assumption is not that practical for most real-world software engineering tasks. This limits the scalability of the data creation pipeline to new, arbitrary codebases.
3. **RL Reward Mechanism:** The reward signal for GRPO is R = speedup^2 for successful optimizations. This non-linear, unbounded reward could be noisy and unstable, disproportionately rewarding simple fixes with massive speedups (e.g., 10x speedup -> 100 reward) over more complex optimizations with modest gains (e.g., 2x speedup -> 4 reward).  The underlying reward signal itself is highly skewed (even GRPO has group normalization), which could bias the model toward low-hanging fruit. Moreover, I have concern on the sparse nature of the reward function when most solutions fail to compile or slower than the baseline.
4. **Minor**: While the paper defines its terms, the methodology in Section 2 can be dense. The introduction of multiple terms (e.g., "PerfCoder Jr.", "plan+code mode", "plan-only mode", "[SUGG/]", "[OPT/]") without in-place explaination makes the pipeline slightly difficult to follow on an initial read.

**Questions:**

1. To improve reproducibility, could the authors please provide more specific details on the GRPO training?

2. In Section 2.1, the notation x^(u,p) is used. It appears the user (u) and promble (p) are simply indexs to identify a unique data point (a specific user's submission to problem p). Is this understanding correct, or is the user identity somehow used as a feature trained in the model?

---

### Official Review · Reviewer_qh5w · 2025-11-01

**Soundness:** 3
**Presentation:** 3
**Contribution:** 2
**Rating:** 4
**Confidence:** 4

**Summary:**

This paper introduces PerfCoder, where LLMs are fine-tuned to produce human-readable optimization strategies and the corresponding optimized code in a single pass. Further alignment with GRPO shows additional improvements. On the PIE dataset, the best 7B shows a 2.5x single-step speedup and is a strong planner for larger models, and also performs better than a collection of open/closed models on code performance optimization.

**Strengths:**

- The optimization pipeline is interpretable, the model includes a suggestion block and an optimization block during generating optimized code, making edits auditable and reusable; Fig.3 makes this easy to understand
- Solid one-shot results with PerfCoder-QC-7B, with 2.5x improvement. Two step procedure also shows good improvements on larger models like Qwen2.5 32B and GPT-5.
- Reconstruct and rebalance the PIE dataset to reduce category bias and target clear optimization endpoints.

**Weaknesses:**

- When comparing against the PIE paper, PerfCoder's "PIE-xxx-HQ" baselines are plain SFT models on the HQ dataset; the PIE paper shows that performance conditioning during training can make the best@1 and best@8 speedups significantly better. The performance-conditioned baselines are not included in PerfCoder's main table.
- Lines 86-87 mention PIE-CodeLlama at 1.89x speedup while Table 2 lists 1.73x.
- PerfCoder says it follows the PIE evaluation protocol, but does not explicitly confirm gem5 usage, which is central to PIE's deterministic evaluation. Clarification is needed for reproducibility.

**Questions:**

- Did the “PIE‑CodeLlama‑HQ / PIE‑Qwen2.5‑Coder‑HQ” baselines include performance‑conditioning at training time or target performance tags at inference (for example 10/10) as in the best PIE results?
- The original PIE paper used a temperature of 0.7 during decoding. Is there a reason to use greedy search in PerfCoder? How would sampling with T=0.7/1.0 affect the results of this paper?
- Is the evaluation of speedups in gem5, or on host hardware? If gem5 is not used, please detail the hardware, OS isolation, compiler, and flags (PIE uses GCC9.3.0 -o3), and how you control variance, so that the results in the paper are reproducible.

---

### Official Review · Reviewer_hkgH · 2025-11-02

**Soundness:** 2
**Presentation:** 3
**Contribution:** 3
**Rating:** 4
**Confidence:** 4

**Summary:**

The paper proposes PerfCoder, a system that trains an LLM to emit a structured optimization plan (comprising strategy categories and short explanations) and then generate the optimized code in a single pass using control tokens. This enables the model to function both as an end-to-end optimizer and as a planner that produces strategies for a larger, frozen "optimizer" LLM to follow. The planner is further fine-tuned with RL. Reported results on PIE show a 7B single-step model achieving a 2.50× average speedup when deployed as a planner. The training data is reconstructed to retain only the final (or global-best) submissions, organized into 15 optimization categories.

**Strengths:**

* Planner-only RL for optimization: The GRPO setup rewards strategies based on measured speedups while keeping the optimizer model frozen. This is my favorite contribution from the paper.

* Strong empirical results: The single-step 7B model achieves a reported 2.50× speedup, and the planner substantially boosts larger models, suggesting that the emitted plans are actionable (but see my notes on the harness).

**Weaknesses:**

* The relationship and differences between PerfCoder and PIE's CoT/rationale require clearer explanation (reason for overall rating = 4)

PerfCoder claims to "generate human-readable optimization strategies tailored to the program and apply them transparently," which closely resembles PIE's approach via reasoning/rationales (CoT-style). To better clarify the comparative value, consider including the following baseline comparisons: (a) a strong free-form CoT (no schema) planner using the same RL, and (b) a retrieval-augmented edit learner (retrieving top-K slow => fast exemplars) without schema, both with and without RL. These comparisons are necessary to determine whether PerfCoder's structured schema and planner-RL approach offers substantial improvements over PIE-style CoT or retrieval methods, rather than simply reframing similar strategies.


* Runtime measurement details are insufficient for a speedup-focused paper (reason for overall rating and soundness score = 2)

The paper does not fully specify the measurement harness (hardware, compiler & flags, warm-ups, aggregation method, etc.). Clear reporting of these elements is critical for program optimization work. Please confirm that you will release the exact runtime harness (scripts, Docker configuration, CPU pinning details) and report variance/confidence intervals. Explicitly document compiler flags and any gcov usage.

**Questions:**

Please see weaknesses. I also had two nits (not important, please don't worry about addressing these)


1. Confining suggestions to 15 categories may help with supervision but constrains the hypothesis space, potentially omitting complex tactics.

2. On a related note, using only the last submission (or global-best replacement when the last is far from optimal) simplifies labeling but discards intermediate trajectories that often contain useful micro-optimizations and diverse tactics.

---

### Note · Authors · 2026-01-05

I have read and agree with the venue's withdrawal policy on behalf of myself and my co-authors.